# IGFBP-6 Network in Chronic Inflammatory Airway Diseases and Lung Tumor Progression

**DOI:** 10.3390/ijms24054804

**Published:** 2023-03-02

**Authors:** Santina Venuto, Anna Rita Daniela Coda, Ruperto González-Pérez, Onofrio Laselva, Doron Tolomeo, Clelia Tiziana Storlazzi, Arcangelo Liso, Massimo Conese

**Affiliations:** 1Department of Medical and Surgical Sciences, University of Foggia, 71122 Foggia, Italy; 2Allergy Department, Hospital Universitario de Canarias, 38320 Tenerife, Spain; 3Severe Asthma Unit, Hospital Universitario de Canarias, 38320 Tenerife, Spain; 4Department of Clinical and Experimental Medicine, University of Foggia, 71122 Foggia, Italy; 5Department of Biosciences, Biotechnology and Environment, University of Bari Aldo Moro, 70125 Bari, Italy

**Keywords:** IGFBP-6, airway diseases, inflammation, lung cancer

## Abstract

The lung is an accomplished organ for gas exchanges and directly faces the external environment, consequently exposing its large epithelial surface. It is also the putative determinant organ for inducing potent immune responses, holding both innate and adaptive immune cells. The maintenance of lung homeostasis requires a crucial balance between inflammation and anti-inflammation factors, and perturbations of this stability are frequently associated with progressive and fatal respiratory diseases. Several data demonstrate the involvement of the insulin-like growth factor (IGF) system and their binding proteins (IGFBPs) in pulmonary growth, as they are specifically expressed in different lung compartments. As we will discuss extensively in the text, IGFs and IGFBPs are implicated in normal pulmonary development but also in the pathogenesis of various airway diseases and lung tumors. Among the known IGFBPs, IGFBP-6 shows an emerging role as a mediator of airway inflammation and tumor-suppressing activity in different lung tumors. In this review, we assess the current state of IGFBP-6’s multiple roles in respiratory diseases, focusing on its function in the inflammation and fibrosis in respiratory tissues, together with its role in controlling different types of lung cancer.

## 1. Introduction

Epithelial–mesenchymal interactions on a proper extracellular matrix play a crucial role in the structural and functional development of the lung [1]. However, the full spectrum of the growth factors involved in this process needs to be completely elucidated. Notably, the lung is the central organ for gas exchanges and presents a large epithelial surface; consequently, it is in contact with the environment, many air pathogens, allergens, and aerosols. It is also a crucial organ for inducing a potent immune response, harboring both innate and adaptive immune cells [2]. Lung homeostasis is based on a critical balance between inflammation and anti-inflammation factors, and, therefore, a comprehensive understanding of the inflammatory mechanisms is essential for treating patients affected by lung inflammation.

Inflammation is a natural defense mechanism for removing dangerous stimuli such as irritants, pathogens, and damaged cells, and, consequently, initiate a recovery process. Through the activation of macrophages and epithelial cells, crucial cells for the innate immune response, acute inflammation lets neutrophils mobilize to repair injuries, helping to confine the injured region [3]. In the presence of unresolved injuries, chronic inflammations can develop, recruiting immune cells from the bloodstream (mainly monocytes and lymphocytes) in order to further amplify the inflammatory response, often determining damage to the tissues. Then, the apoptotic process and consequent clearance of the initiated inflammatory cells will activate the tissue healing process to repair the damaged tissue, eventually leading to fibrosis [4]. Following the exposure to different factors that affect the inflammatory response, such as diseases, stress, or seasonal changes, various signaling pathways are affected by changes in the tissue microenvironment, and, consequently, immune cells are triggered to reach the injured tissues. Consequently, the interaction between epithelial cells and neutrophils provides communication during inflammatory responses. The last ones have been observed in various diseases, and their over-regulation that occurs with advancing age is documented as one of the main actions that lead to tissue damage [3]. 

Inflammation in the lung is the immune system’s natural response to damage. It orchestrates the removal of negative stimuli such as irritants, pathogens, and damaged cells, and initiates the healing process. Pulmonary inflammation could be acute and chronic, and these responses are both established in different respiratory diseases, such as asthma and cystic fibrosis (CF) [2]. 

Several pieces of evidence demonstrate the involvement of the insulin-like growth factors (IGFs) and their binding proteins (IGFBPs) in normal pulmonary development, as they are expressed in different lung compartments: *IGF-1* and *IGF-2* mRNA have been found in lung tissue [5], and IGF-1 is also produced in fetal lung tissue or mesenchymal cells, especially those near the airway epithelium [6]. Type 2 IGF receptors are similarly expressed in lung tissue [7], as well as mRNAs encoding type 1 and type 2 IGF receptors [6], and IGFBP-1 to -6 [6,8]. IGF-1, IGF-2, and the IGF-1 receptor (IGF-1R) are involved in the physiologic development of the pulmonary tissue and also in the pathogenesis of smooth muscle tumors [9]. The mitogenic and metabolic activities of IGFs are modulated by a family of six high-affinity IGFBPs. Among the IGFBPs, IGFBP-6 is unique due to its preferential binding to IGF-2, and it is strongly associated with cellular and immune system processes and their biological regulation. Regarding pulmonary expression, *IGFBP-6* mRNA levels are significantly increased in perinatal and fetal rat lungs [5]. Then, IGFBP-6 is found at the epithelial level of human bronchial organ cultures and primary cultures of human bronchial epithelial (HBE) cells, and it is also involved in inhibiting their growth [10]. 

Normal human inflammatory airway cells produce a plethora of potent neurohormones and IGFBPs that modulate the bioavailability of IGF-1 in the lung [11]. In this inflammatory scenario, IGFBP-6 is differentially expressed in the lung, in association with the suppression of allergic airway inflammation, and bronchial biopsies of asthmatic subjects [12,13], and was recently reported as an innovative regulator of the inflammatory response in CF airway cells [14]. It is also reported that inflammatory cells of the normal human airway express IGF-1 and IGFBPs, and that IGFBP-6 is made by alveolar macrophages’ precursors, peripheral blood monocytes [11]. 

The lung has developed a finely regulated multi-level system able to respond to continuously inhaled and aspirated material, orchestrated as a pulmonary host defense. However, the breach of mechanical and biochemical barriers, phagocytic scavengers, and humoral immune reactions can lead to the development of respiratory tract infections [15]. Keratinocyte growth factor (KGF) is an epithelial mitogen that enhances the pulmonary host defense and protects the lungs from a multiplicity of insults. Interestingly, KGF treatment was correlated with increased levels of IGFBP-6 in the bronchoalveolar lavage [16]. Furthermore, a fluid and microarray analysis of implantation and explantation samples from recovery patients revealed elevated IGF-2, IGFBP-4, and IGFBP-6 levels [16]. All these findings confirm the IGFBP-6 role as a mediator of airway inflammation, supporting previously reported data [17]. 

In another scenario, alveolar macrophages exert a focal role in sarcoidosis, releasing effective profibrogenic molecules such as IGF-1. Moreover, IGFBPs contribute to the fibrogenic process in patients with well-known stage III sarcoidosis with implications in the disease progression [18]. These findings highlight the IGFBPs’ role in the pathogenesis of pulmonary fibrosis, indicating that the multifaced IGFBP network is an important element for the establishment of a fibroproliferative response to injury in the lung [19]. The IGFs/IGF-Rs/IGFBPs system and its role in the lungs are depicted in Figure 1.

In this review, we focused on describing the multiple roles of IGFBP-6 in respiratory diseases, with particular attention paid to those that are related to its function as a mediator of inflammation and fibrosis in the lung and bronchial tissues, together with its role in controlling different forms of lung cancer.

## 2. IGFs, IGFRs, and IGFBPs

The IGF family comprises a total of 14 active peptides, including insulin, IGF-1, IGF-2, relaxin 1–3, Leydig cell-specific insulin-like peptide (LeyIL), early placenta insulin-like peptide (EPIL), insulin-like peptides 5 and 6, and IGF-like (IGFL) 1–4 [20,21]. While the critical role that the IGF signaling system plays in cellular energy metabolism, growth, and development [22,23] is well-known, other paracrine and endocrine actions have been discovered for IGF-1 and IGF-2, notably in inflammation [24,25]. IGFs binding to different receptors mediate their actions that are finely regulated by binding proteins. IGF receptors comprises IGF type 1 (IGF-1R), IGF type 2 receptor (IGF-2R), insulin receptor type A (IR-A), insulin receptor type B (IR-B), and hybrid insulin/IGF-I receptor [26]. A receptor for IGFL1 has been found to be expressed by T cells and monocytes [27]. The binding of IGFs to their receptors, especially IGF-1R, on specific target cells activates several intracellular signal transduction pathways, in particular, the PI3K/Akt and Ras/MAP kinase pathways, but also, other accessory pathways, kinases, adaptor proteins, and scaffolds have been implicated in modulating IGF/IGFR signaling [28]. Through these complex interactions, IGFs promote cellular growth, survival, proliferation, and differentiation [29]. At the extracellular environment level, IGFs are bound to six specific, high-affinity IGFBPs that serve as carrier proteins but also regulate the turnover of circulating IGFs together with their transport, cellular distribution, and many cell activities. Locally expressed IGFBPs can inhibit and/or potentiate IGF activities [30]. By sequestering IGFs away from the type 1 IGF receptor, they may inhibit mitogenesis, differentiation, survival, and other IGF-stimulated events. Moreover, IGFBP proteolysis can reverse this inhibition or generate IGFBP fragments with novel bioactivity [30]. Alternatively, the IGFBP interface with the cell or matrix components may increase the IGF pool near their receptor, enhancing IGF activity [31].

However, the bioactivity of IGFs is not only dependent on their interaction with IGFRI. Indeed, the IGFBP family in the local cellular environment can also influence the IGF bioavailability either inhibiting or enhancing IGF actions, depending on the IGFBP complementation [31]. Considering this, the physiologic and pathologic contexts in which functionally distinct IGFBP members are expressed can either restrict or promote the availability of IGFs to the IGFRs, thus finely regulating the extent of IGF signaling [30,31]. 

Immunity and growth are independently ruled by signaling pathways inducted by conserved endocrine factors such as cytokines and hormones. The interplay between such pathways allows the regulation of resource allocation, depending on growth and immune status [32]. Therefore, it is possible to hypothesize that, as a central regulator of vertebrate development and metabolism, the IGF axis is a prime candidate to be targeted during an immune response, as it participates in modifying the availability of IGF to IGF-1R or other receptors [32]. The relationship between the immune system and the IGF/IGFR/IGFBP axis has yet to be fully unveiled. So far, we know that lymphocytes and granulocytes express components of the IGF system, implying a role in the differentiation, growth, or functions of the immune system [24]. Furthermore, in response to the treatment of mice and cells by proinflammatory cytokines (including IL-1α, IL-1β, and TNF-α), the IGFBP family members IGFBP-1 and IGFBP-6 are upregulated as a result [33,34].

IGFBPs also have IGF-independent cellular effects, including cell migration, fibroblast proliferation, apoptosis, and angiogenesis [17]. We have recently highlighted a regulatory role for IGFBP-6 in the immune system. IGFBP-6 is upregulated by hyperthermia in monocyte-derived dendritic cells (DCs) and stimulates the chemotaxis of T cells, monocytes, and neutrophils (PMN), as well as intensifies the PMN oxidative burst [35,36,37,38]. Overall, these results imply that IGFBP-6 may have an important role in innate immunity and in the elicitation of adaptive immune cells, as well as in the outcome of these responses, i.e., inflammation resolution vs. fibrosis [39]. 

## 3. IGFBP-6 Expression in the Lung

Several pieces of evidence indicate the role of the IGF and IGFBP signaling pathways in lung development and diseases, including inflammation, congenital disorders, tumor, and fibrosis. The IGF network also controls the maturation and differentiation of several types of lung cells, counting airway basal cells, alveolar epithelial cells, and fibroblasts [40]. 

IGFBPs are expressed and delivered in spatially and temporally different compartments of the developing lung and are important paracrine regulators of IGF actions in the lung [41]. Particularly, IGFBPs were identified in lung tissues, and cultured fetal lung cells were able to product IGFBPs; thus, they have a possible role in lung development. Among the six IGFBPs, IGFBP-6 is localized to the large cartilaginous airways in the adult and it was also detected in an embryonic lung, and in a day-18 late gestational mouse fetus [41,42], with different sites of synthesis and actions [43]. 

In a more recent study, the developmental patterns of *IGFBPs*’ mRNA abundance were measured in both perinatal and explant cultures of fetal rat lung, demonstrating that *IGFBP-6* mRNA levels significantly increased and may also play a role in keeping the development status in the adult lung. Interestingly, dexamethasone and other glucocorticoids may concern the developmental regulation of the IGF system expression, as they increased the abundance of *IGFBP-6* mRNA in the explant cultures [5]. As IGFBP-6 has a greater affinity for IGF-2 than IGF-1, exerting an inhibitory effect on IGF-2 [26], the elevated expression of IGFBP-6 in the mature lung may thus be necessary for the inhibition of IGF-2 actions at a developmental stage [5].

## 4. IGFBP-6, Fibrosis and Respiratory Diseases

IGFBP-6 is differentially expressed in the bronchial biopsies of asthmatic subjects [12]. As it is a critical regulator of IGF bioavailability, it was demonstrated that the basal epithelial layer of human bronchial organ cultures expresses IGFBP-6 [10], where it correlates with the basal cell subpopulation marker cytokeratins 14 [44]. Remarkably, basal cells are an assorted population containing epithelial stem cells that characterize the conducting airway’s pseudostratified epithelium [45].

Starting from its expression in the main respiratory system tissues, several studies have shown that IGFBP-6 is involved in the proliferation and development of lung and bronchial cells since fetal development as well as in different pathologies and disorders (Table 1), as extensively discussed in the following paragraphs. Figure 2 presents the major features of IGFBP-6 involvement in respiratory diseases.

Several IGFBP superfamily members play a critical role in the initiation and maintenance of lung fibrosis [46]. The effects of IGFs are difficult to understand because the cellular responses they generate depend on their receptors’ expression, the tissue-specific type of IGFBPs, and the balance of IGFBP synthesis, post-translational modifications, and degradation [47]. As already reported, IGFBP-6 exerts a central role in tissue remodeling, fibrosis, and immunity [39], and several studies showed that IGFBP-6 is implicated in fibrotic lung pathologies, as discussed below. 

**Table 1 ijms-24-04804-t001:** IGFBP-6 expression in respiratory diseases.

Diseases	IGFBP-6 Function	IGFBP-6 Expression/Levels	Ref.
Asthma	Asthma pathophysiology, cell growth, and proliferation	Differentially expressed in two microarrays studies using bronchial tissues of asthmatic and control subjects	[12]
Bronchial tissue inflammation	Pathophysiologic process active in the asthmatic lung	Significantly upregulated (4.03-fold change; *p* = 0.001) in bronchial tissues obtained from asthmatic patients after inhaled corticosteroids	[48]
Allergic airway inflammation	Suppression of allergic airway inflammation	Significantly increased in asthmatic mice (1.52-fold change) after anti-allergic treatment	[13]
Acute mountain sickness (AMS)	Predicting AMS susceptibility	Significantly lower in AMS-susceptible individuals (37,318.99 ± 23,493.11 pg/mL and 25,665.38 ± 25,691.29 pg/mL, respectively; *p* = 0.04).	[49]
Lymphangioleiomyomatosis (LAM)	Proliferation of LAM cells, associated with of spindle-shaped LAM cells	Upregulated (1.82, biopsies; 1.57, explants; *p* < 0.05) in patients compared to controls.	[9,50]
Cystic fibrosis (CF)	Mediator of airway inflammation	IGFBP-6 mRNA and protein levels are both upregulated in bronchial epithelial F508del-CFTR CFBE cells lines and primary nasal epithelial cells (HNE) from three CF patients bearing the most common CF-causing mutation (F508del)	[14]

### 4.1. IGFBP-6 in Asthma Progression

Asthma is a heterogeneous disease, usually characterized by chronic airway inflammation. It is defined by the history of respiratory symptoms, such as wheezing, shortness of breath, chest tightness, and cough, that vary over time and in intensity, together with a variable expiratory airflow limitation [51]. Asthma is characterized by the chronic inflammation of the pulmonary airways, comprising several pheno-endotypes with distinct clinical features, pathobiological mechanisms, and biomarker expression [52,53]. Its remarkable features are airway hyperresponsiveness (AHR), persistent airway inflammation, and airway remodeling [54]. 

Concerning the pathobiology of asthma, inflammation of the lung is primarily characterized by a dichotomy of a preponderant (>50%) type 2-high response—including eosinophilic, allergic, and non-allergic asthma—and a type 2-low response involving a neutrophilic and pauci-granulocytic asthma, if there is no evidence of elevated sputum eosinophils or neutrophils, and if treatments aimed at suppressing eosinophils and neutrophils are ineffective in controlling the symptoms [55]. In this last endotype, elevated levels of IFN-γ and other cytokines released from Th1, Th17, or type 3 innate lymphoid cells are observed [55,56,57]. In most cases, the asthmatic phenotype with a neutrophilic inflammation is associated with the Th17 pathways, increased IL-8 production, and stronger activation of innate immune mechanisms [58].

In this regard, the pathogenesis of the allergic inflammation of the airways is often due to the excessive activation of Th2 cells together with the deficient suppression of regulatory T cells (Tregs) [13]. Thus, biological agents targeting specific immune mediators such as type 2 cytokines (IL-4, IL-5, and IL-13) or immunoglobulin E (IgE) have emerged as a safe and effective tailored therapy for severe asthma [59,60]. Wang et al. showed that IGF signaling evolves important functions in asthma physiopathology at different stages, including type 2 high inflammation, eosinophilia, mucus production, bronchial hyperresponsiveness, and lung remodeling [40]. Interestingly, several studies have shown that mesenchymal stem cells (MSCs) can recover allergic airway inflammation in asthmatic mice [61,62]. 

The immune suppression mechanism of MSCs in allergic airway disorders is strongly related to Treg expansion and upregulated levels of transforming growth factor-β (TGF-β), and interleukin- (IL-) 10. Many studies also confirmed that MSC-driven immunomodulation is mediated by the reduction of proinflammatory Th1 responses with a concomitant rebalancing of the Th1/Th2 ratio on Th2 [63]. As widely demonstrated, all these factors are related to IGFBP-6, which is an important mediator of chemotaxis and the immune response [39]. 

IGFBP-6 is involved in the pathophysiology of asthma, as demonstrated in different genome-wide association studies, the candidate genes approach, and the genomic expression and linkage analysis which identified over 300 genes associable with asthma pathophysiology, including IGFBP-6 [12]. Then, IGFBP-6 may be involved in the amelioration of allergic airway inflammation, as its expression was significantly increased in the lung of asthmatic mice following the treatment with adipose-stem-cell-derived extracellular vesicles, a condition normally associated with the suppression of allergic airway inflammation [13]. All this information may be used to shed light on the basic biology and disease pathogenesis, which will conclusively promote the development of personalized therapies targeting the cause of the underlying disease. Recent studies suggested that inflammation and cytokines are closely related to acute mountain sickness (AMS) [64], an inflammatory condition originated by a hypoxia-induced hypoxemia reaction with the consequent release of inflammatory mediators that led to over-perfusion and vasodilatation, contributing to the increase of the capillary pressure [49]. Patients with asthma could develop symptoms of AMS. AMS-resistant individuals have a greater capability to control anti-inflammation damage than AMS-susceptible individuals, as demonstrated by the comparison of their plasma cytokine profiles at low altitudes. IGFBP-6 levels were significantly lower in AMS-susceptible individuals; thus, it may be implicated in predicting AMS susceptibility in low-altitude conditions [49].

In asthmatic subjects, the primary site for airway inflammation and remodeling is the bronchial tissue [65]. Specifically, airway remodeling in asthma is pathologically characterized by mucosal hypersecretion and goblet cell metaplasia, a thickening of the subepithelial basement membrane due to an increased deposition of extracellular matrix proteins, airway smooth muscle hyperplasia, and angiogenesis [66,67]. The gene expression profile obtained from asthmatic subjects’ bronchial tissues before and after the treatment with inhaled corticosteroids (ICS) was used to identify genes associated with asthma pathogenesis, together with the original susceptibility genes. Comparing the subjects with allergic asthma and healthy controls in their bronchial biopsies before and following an inhaled corticosteroid (ICS) therapy, 74 genes were found to be differentially expressed. Among differentially expressed genes that control cell growth and proliferation, there is also *IGFBP-6*, which presented a lower expression in asthmatic patients as compared with healthy controls that was recovered by ICS (Figure 2), providing insights into the pathophysiologic process active in the asthmatic lung and clarifying one of its possible roles in the natural history of asthma [48]. 

To summarize, IGFBP-6 levels are significantly higher in asthma after ICS therapy. Therefore, IGFBP-6 may be involved in the amelioration of allergic airway inflammation; thus, enhancing the IGFBP-6 expression might lead to the improvement of current therapies.

### 4.2. Idiopathic Pulmonary Fibrosis (IPF) and Lymphangioleiomyomatosis (LAM)

IGF and IGFBP expression is deregulated in IPF, a form of chronic interstitial pneumonia depicted by fibrotic alterations heading to alveolar destruction and ongoing obstructive lung disease [68]. Events that possibly contribute to the final lung fibrosis, such as fibroblast activation and trans-differentiation to a myofibroblast phenotype, increased ECM production and epithelial–mesenchymal transition (EMT), and decreased ECM degradation, designating the IGFBPs’ involvement in the beginning and progression of the fibrosis process [46].

LAM is characterized by the proliferation of abnormal smooth muscle cells (LAM cells), as well as the formation of nodules in the pulmonary interstitium and multiple cysts throughout the lungs [69] (Figure 2). It is strongly suggested that the IGF system is involved in the proliferation of LAM cells [9]. Evaluating the expression of IGFs, IGF1R, and IGFBPs in the lungs of patients with LAM, Valencia et al. demonstrated that IGFBP-6 is localized and can be synthesized by LAM cells and can modulate the effects of the IGFs on LAM cell proliferation [9]. IGFBP-6 was associated with spindle-shaped LAM cells, a typical LAM neoplastic cell type, and appeared to be involved in their development [9,50].

The patterns of IGFBP-6 localization and expression in LAM strongly suggest that it is involved in the proliferation of LAM cells, modulating the IGF effect on them. Blocking its expression could be revealed as an efficient hormone therapy to regulate the expression of IGFBPs. 

### 4.3. Cystic Fibrosis (CF)

CF is a common life-limiting genetic disorder, caused by the mutation of a gene that encodes a chloride-conducting transmembrane channel called the CF transmembrane conductance regulator (*CFTR*), resulting in the failure of chloride secretion and sodium hyperabsorption with the consequent alteration of the anion transport and mucociliary clearance in the airways [70]. This functional failure leads to several conditions harmful to the lung, such as viscous mucus retention at the epithelial surface, chronic infection, and, subsequently, a local airway inflammation [70,71]. As a result, CF patients are exposed to the contraction of bacterial lung infections with opportunistic pathogens, associated with chronic inflammation in the CF lung, whose hallmarks are increased levels of neutrophil attraction by chemokines [72].

The involvement and role that IGF superfamily members and IGFBPs may have in CF lung disease are completely unknown. We have recently highlighted that IGFBP-6 may play a direct role in CF-associated inflammation. The basal IGFBP-6 mRNA and protein levels are both upregulated in the bronchial epithelial F508del-CFTR CFBE cells lines and primary nasal epithelial cells (HNE) from three CF patients bearing the most common CF-causing mutation (*F508del*). Moreover, we found that the IGFBP-6 expression was further increased after infection/inflammation stimulation in both the CFBE cell line and HNE cultures. Treatment with a clinically approved anti-inflammatory drug (dimethyl fumarate) significantly reduced the *IGFBP-6* mRNA levels in the CFBE cell lines and HNE cultures, suggesting its role in CF inflammation. Moreover, it was demonstrated that IGFBP-6 downregulated the level of pro-inflammatory cytokines in both the CFBE and primary nasal epithelial cells. Therefore, we hypothesized that IGFBP-6 may have an autocrine regulation on CF inflammation by downregulating the pro-inflammatory cytokines to attenuate the airway inflammation under an infection/inflammation condition (Figure 2). Lastly, IGFBP-6 treatment did not affect the wild-type and rescued F508del-CFTR protein expression and function, demonstrating its specific role in inflammation regulation [14]. 

## 5. IGFBP-6 Controls the Proliferation of Normal and Neoplastic Epithelial Cells

IGFs are expressed in the lung and are potent mitogens of normal and neoplastic cells originating from the bronchial epithelium. Remarkably, IGFBP-6 is a crucial regulator of IGF bioavailability and was found to be abundantly expressed in the basal epithelial cells of bronchial organ cultures and in primary cultures of HBE cells. 

### 5.1. IGFBP-6 Role in the Proliferation of Epithelial Cells

It is already known that functional retinoid response elements exist in the IGFBP-6 promoter [73]. Retinoids, the biological and synthetic derivatives of vitamin A, are usually used in the treatment and prevention of cancer. In a study for all-trans-retinoic acid (t-RA) target genes linked to growth suppression in BEAS-2B human bronchial epithelial cells, it was found that RA deregulated the expression of different genes, with a prominent induction of IGFBP-6, identified as a direct RA target [74]. All these findings are consistent with a potential role for IGFBP-6 as a direct RA target in human bronchial epithelial cells. 

t-RA is a potent inhibitor of HBE cell development and it is already known that retinoid receptors directly trigger the expression of genes encompassing retinoid response elements in their gene promoters. IGFBP-6 is expressed in HBE cells, increased in abundance with retinoid treatment exerting an interesting growth inhibitory effect on these bronchial epithelial cells [10].

We have recently demonstrated that IGFBP-6 is able to induce mesenchymal stromal cell differentiation associated with an upregulation of cancer-associated fibroblast (CAF) markers, which plays a central role in the activation of the SHH pathway throughout the fibrotic process [75]. In the context of airway inflammation and diseases, gene expression and secretome analyses of CAFs and normal lung-associated fibroblasts (NAFs) disclosed the differential abundance of IGFs and IGFBPs, which promoted or inhibited, respectively, IGF1R signaling. Conversely, drug sensitivity was decreased by the recombinant IGFs or conditioned medium from CAFs in which IGFBP-6 was silenced [76]. Moreover, the Smoothened (Smo) inhibitors of the Hedgehog (HH) pathway are activated in lung cancers and IGFBP-6 is differentially expressed in human normal-malignant lung tissue arrays [77,78]. In a study based on lung cancer cell lines, transgenic and transplantable murine lung cancer models, and a human normal-malignant lung tissue array, the sensitivity to the Smo inhibitor cyclopamine correlated with low IGFBP-6 [78].

Starting from this data, in the next paragraph, we will discuss IGFBP-6’s role in lung cancer pathogenesis and progression.

### 5.2. IGFBP-6 Role in Lung Cancer

According to the World Health Organization, lung cancer is the second most common cancer type and the leading cause of cancer mortality worldwide [79]. Histologically, two main types of lung cancer can be distinguished: non-small-cell lung cancer (NSCLC) and small-cell lung cancer (SCLC) [80].

As with other cancer types, the growth factor receptor tyrosine kinase signaling also plays a crucial role in lung cancer initiation and progression. Indeed, the activation of tyrosine kinase receptors, such as IGF-1R, can lead to the downstream signal transduction pathways promoting cell proliferation and survival [81]. Thus, these receptors have become particularly relevant targets for drug treatment [22].

IGF-1 and IGF-2 are mitogenic peptides in various cells, including those of lung cancer [82]. Particularly, SCLC cell lines were shown to be responsive to IGF-1 [83]. Emerging studies have also disentangled the role of the IGF axis in NSCLC [84]. In detail, it has been demonstrated that IGF-1R signaling can contribute to malignant transformation and cancer growth and that the upregulation of IGF-1R and IGF-2 as well as the deregulation of their downstream signaling molecules increase the risk of cancer development and invasiveness [84]. *IGF-1R* mRNA and protein expression are increased in the serum from NSCLC patients, while IGF-1R downregulation in mice decreases tumor growth, proliferation, and vascularization [85]. Therefore, IGF-1R could be a potential biomarker of the drug response and clinical progression in NSCLC patients, as it was demonstrated to act as a promoter of metastasis and tumor progression in the lung tumor microenvironment [85]. Consequently, therapeutic strategies targeting the IGF axis, such as IGF-1R inhibition by OSI-906 [81], brigatinib [86], and osimertinib [76], have shown promising efficacy and could have an important anti-tumoral potential in lung cancer. 

IGFBPs function as modulators of IGF signaling and have been investigated for their role as possible biomarkers in lung cancer [87,88]. Notably, IGFBP-3 has been well-documented to show an inverse correlation with lung cancer [89], considering the lower levels of this IGFBP in the serum of patients vs. controls, as observed in several studies [90], suggesting it as a potential biomarker [87]. Similarly, IGFBP-7 was shown to be downregulated in the cancer tissues and serum from NSCLC patients, suggesting, also for this IGFBP, a role as a diagnostic biomarker [90]. Notably, both genes encoding these two proteins were found inactivated by DNA-hypermethylation in lung cancer cell lines and primary lung tumors, and their downregulation was associated with cisplatin-resistance in NSCLC [91,92,93,94]. However, the role of IGFBP-7 in lung cancer is not very clear, as other studies indicate that high serum levels of this protein correlated with a positive nodal status [95], and that the gene over-expression was markedly increased in AZD9291-resistant cell lines and patients [96]. Another recent study demonstrated that enhanced IGFBP-7 expression promoted resistance to epidermal growth factor receptor tyrosine kinase inhibitors in lung cancer cells by mediating the IGF-1R pathway [97]. Hence, the role of IGFBP-7 in lung cancer is somewhat controversial and requires further investigation. 

Concerning other IGFBPs, IGFBP-5 was reported in only one study as a possible biomarker for the progression and outcome in lung cancer [95]. IGFBP-4 levels in the serum of lung patients were higher than those of healthy controls [98], suggesting a potential role as a biomarker in lung cancer with an adverse association with the prognosis [99]. Similarly, the IGFBP-2 expression level in lung cancer patients was found to be significantly higher than that in controls and associated with an advanced tumor stage. Furthermore, the over-expressed IGFBP-2 could predict chemoresistance that could be reversed by Trichostatin A by enhancing autophagy in vitro [100].

Focusing on IGFBP-6, this protein was firstly identified as secreted by a normal human lung fibroblast cell line (He[39]L) [101]. Then, *IGFBP-6*-specific transcripts were found to be expressed in NSCLC cell lines and primary tumors [102]. The importance of this gene in lung tumors was underlined in a gene expression profile study on lung adenoma vs. adenocarcinoma progression in mice, showing *IGFBP-6* as a candidate gene for lung tumor progression, as its downregulation may affect lung tumor progression [103]. Moreover, by analyzing the distinct expression patterns and prognostic values of IGFBP family members in patients with NSCLC with bioinformatics tools, *IGFBP-6* was downregulated in tumor samples as a result [88]. These results were confirmed by detecting a low IGFBP-6 expression level in the serum from NSCLC patients by an antibody array and subsequent ELISA validation [104]. Other pieces of evidence demonstrated that IGFBP-6 inhibited NSCLC cell growth through the activation of programmed cell death [10]. This effect is exerted either directly or indirectly through the activation of Semaphorin B, which acts as a tumor suppressor gene by inducing apoptosis [105]. Thus, in both the adenocarcinoma and squamous cell carcinoma subtypes of NSCLC, the IGFBP-6 expression has been found to be lower compared with healthy tissues, resulting in the decreased function of Semaphorin 3B (Figure 2).

In summary, due to the high variability of the IGFBP action in the IGF network, it is possible to subgroup IGFBP genes into two categories: (i) oncogenes, whose products are upregulated in lung cancer, i.e., IGFBP-2/-4/-5; and (ii) tumor suppressors, that are downregulated in lung cancer, i.e., IGFBP-3/-6. However, several molecules involved in cancer may actually have a dual role as oncogenes and tumor suppressors, even in the same cancer type. Among them, IGFBP-7 seem to have such a behavior in lung cancer. Notably, its transcriptional level was decreased in cisplatin-resistant human cancer cell lines and NSCLC xenografts, while a high serum level correlated with a positive metastatic status or after an acquired EGFR-TKI resistance. Despite the fact that there is no evident correlation between each single IGFBP expression pattern and lung cancer, it could be hypothesized that the possible differences in published studies concerning IGFBPs are probably due to the presence of metastases and/or the treatment of the tumor with cisplatin or other inhibitors.

Altogether, these data confirmed that IGFBP-6 has a promising tumor suppressor role in lung tumors, highlighting the need for further investigations to determine if this protein could be used as a novel biomarker, paving the way for new targeted therapies that aim to increase its activity as a potential tumor suppressor.

## 6. Conclusions

Inflammation is a crucial part of the response to injury or infection. However, chronic inflammation can lead to a host of conditions including fibrosis, cancer, and autoimmune disorders. IGFBP-6 plays several complex roles as a mediator of airway inflammation, participating in the pathogenesis of asthma, fibrosis, as well as lung cancer. IGFBP-6 expression levels can play a significant role in determining its effects on chronic inflammation, also determined by the cell type that expresses it. Additionally, IGFBP-6 can have both autocrine effects (acting on the cells by which it is secreted) as well as paracrine effects (acting on nearby cells).

A better understanding of the multiple roles of IGFBP-6 could help the fine-tuning of the inflammatory response and the creation of a favorable environment for the resolution phase, which is critical for preventing chronic inflammation-related diseases.

## Figures and Tables

**Figure 1 ijms-24-04804-f001:**
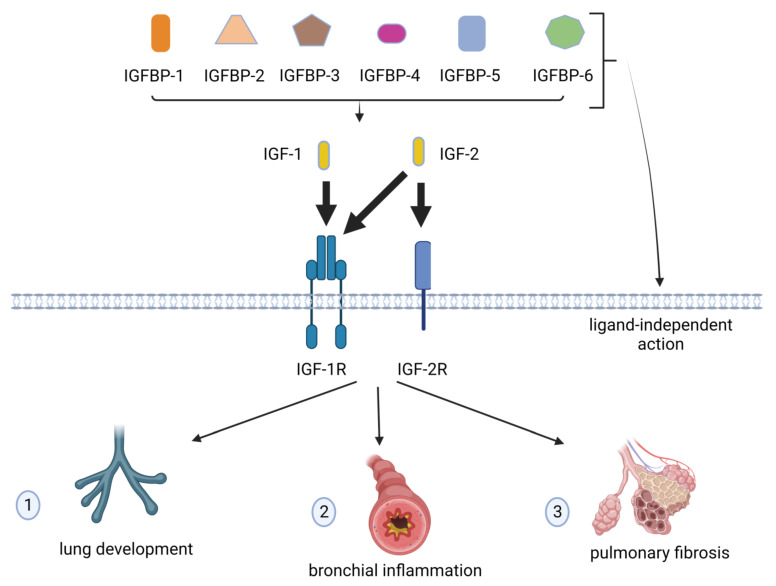
The insulin-like growth factor (IGF) system. Two IGF ligands (IGF-1 and -2), two receptors (IGF-1 receptor (IGF-1R) and type-2 IGF receptor (IGF-2R)), and six forms of IGF-binding proteins (IGFBPs) are shown. IGF-1 and IGF-2 bind to IGF-IR with high affinity, whereas IGF-2 can also bind to IGF-2R. IGFBPs can act by enhancing or inhibiting IGF actions as well as determining ligand-independent effects. In the respiratory system, the IGF/IGF-R/IGFBP system is involved in: (1) lung development, (2) bronchial inflammation, and (3) pulmonary fibrosis. Not all the molecules involved in this system are shown; see text for further details.

**Figure 2 ijms-24-04804-f002:**
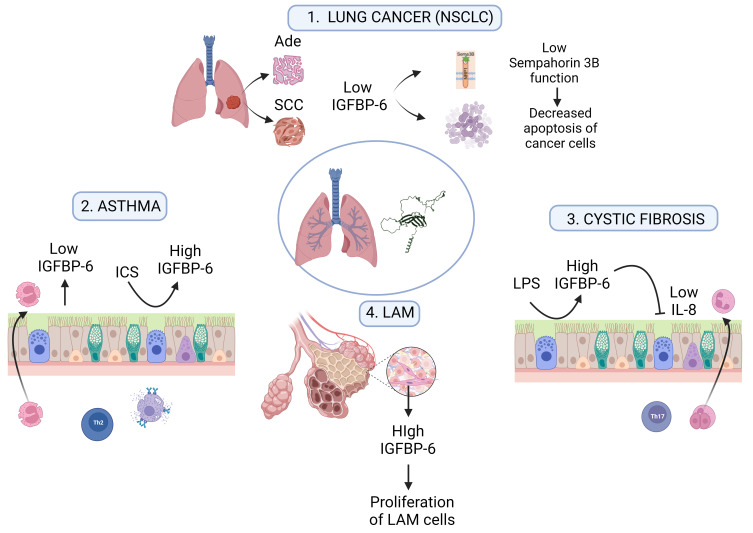
Roles of IGFBP-6 in respiratory diseases. (1) In NSCLC, adenocarcinoma (Ade), and squamous cell carcinoma (SCC) subtypes, IGFBP-6 expression is lower compared with healthy tissues, resulting in decreased function of Semaphorin 3B that in turn acts as a tumor suppressor gene by apoptosis induction. (2) Asthmatic airways, characterized by Th2 cell infiltration, eosinophil recruitment, and mast-cell activation, show low production of IGFBP-6 which is recovered by ICS (inhaled corticosteroid) treatment. (3) Cystic fibrosis airways display a Th17 dysregulation with heightened neutrophil recruitment. Bacterial lipopolysaccharide (LPS) upregulates IGFBP-6 which, in turn, downregulates IL-1β, IL-6, and TNF-α secretion. (4) Pulmonary lymphangioleiomyomatosis (LAM) is depicted by abnormal muscle cell proliferation (LAM cells) as well as the formation of multiple cysts throughout the lungs. High IGFBP-6 expression has been shown in LAM cells, thus contributing to their excessive growth.

## Data Availability

Data sharing is not applicable to this article.

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
