# Peer review of "IGFBP-6 Network in Chronic Inflammatory Airway Diseases and Lung Tumor Progression"

_ijms, 2023, doi:10.3390/ijms24054804_

Round 1
Reviewer 1 Report
The article is consistent within itself. The references are relevant and recent. The cited sources are referenced correctly. Appropriate and key studies are included. The paper is comprehensive, the flow is logical and the data is presented critically.
However, there are some specific comments on weaknesses of the article and what could be improved:
Major points - none
Minor points
1. I would suggest to authors to re-consider the subheadings in their manuscript, because the structure is not very good. For example, section 4. IGFBP-6 and respiratory diseases contains 4.1. asthma, then 4.2. fibrosis and respiratory disease, and then 4.2.1. ... diseases. Asthma should be in paralel with the other diseases.
Also, section 5 and 6 are related, thus, they can be combined in one common - 5, and to use 5.1. and 5.2. as subheadings.
2. Table 1 - third column is not informative - IGFBP-6 levels - this should be named "expression", and if the expression is different - to cite the differences and the factors for that; besides, if the name remains "levels", some numbers should be cited. It can be also named "expression/levels", but the main thing is that this column should contain much more information than provided in the draft.
3. This part needs a revision in the end: "Focusing on IGFBP-6, this protein was first identified as secreted by a normal human fibroblast cell line, He[39]L [94]."
Author Response
The article is consistent within itself. The references are relevant and recent. The cited sources are referenced correctly. Appropriate and key studies are included. The paper is comprehensive, the flow is logical and the data is presented critically.
However, there are some specific comments on weaknesses of the article and what could be improved:
Minor points
Q1. I would suggest to authors to re-consider the subheadings in their manuscript, because the structure is not very good. For example, section 4. IGFBP-6 and respiratory diseases contains 4.1. asthma, then 4.2. fibrosis and respiratory disease, and then 4.2.1. ... diseases. Asthma should be in paralel with the other diseases.
A1. We have moved the asthma paragraph after the Section named “IGFBP-6, fibrosis and respiratory diseases” (now section 4) and renamed it 4.1. The other two paragraphs on diseases are now 4.2. and 4.3, in parallel as suggested.
Also, section 5 and 6 are related, thus, they can be combined in one common - 5, and to use 5.1. and 5.2. as subheadings.
It has been changed according to the Reviewer’s suggestion.
Q2. Table 1 - third column is not informative - IGFBP-6 levels - this should be named "expression", and if the expression is different - to cite the differences and the factors for that; besides, if the name remains "levels", some numbers should be cited. It can be also named "expression/levels", but the main thing is that this column should contain much more information than provided in the draft.
A2. According with the Reviewer’s request, we have renamed the 3rd column “Expression/levels” and included additional information.
Q3. This part needs a revision in the end: "Focusing on IGFBP-6, this protein was first identified as secreted by a normal human fibroblast cell line, He[39]L [94]."
A3. “He[39]L” is the cell line name, it is not an error. However, in the sake of clarity we have now put the name in brackets in the text.
Reviewer 2 Report
The manuscript of Venuto and co-authors is well-written and summarizes the role of binding proteins of IGFs on acute and chronic lung diseases and lung cancer. This review analyzes the literature well. The conclusions correspond to the tasks set. The figure and table complete the literature analysis.
Only one recommendation: It has been reported that IGFBPs are also overexpressed in bronchoalveolar lavage fluid from patients with sarcoidosis. Authors should add the role of IGFBPs in lung sarcoidosis.
Author Response
The manuscript of Venuto and co-authors is well-written and summarizes the role of binding proteins of IGFs on acute and chronic lung diseases and lung cancer. This review analyzes the literature well. The conclusions correspond to the tasks set. The figure and table complete the literature analysis.
Q1. Only one recommendation: It has been reported that IGFBPs are also overexpressed in bronchoalveolar lavage fluid from patients with sarcoidosis. Authors should add the role of IGFBPs in lung sarcoidosis.
A1. We have added a paragraph at the end of the Introduction Section (highlighted in yellow) and two references on sarcoidosis (Allen J.T., Am J Respir Cell Mol Biol, 1998 and Allen J.T., Am J Respir Cell Mol Biol, 1999 (yellow marked in the text), [18] and [19], respectively.
Reviewer 3 Report
I have read the review article by Venuto et al with great interest. The authors summarise the knowledge on the role of IGFBP-6 in lung diseases.
Comments:
· Table 1. Asthma. Please, clarify this. Currently, the role of IGFBP-6 is confusing.
· Figure 1. IGFBP-6 blocks IL1β on the picture, but the description says IL1α. Please, clarify.
· 4.1 Please, clarify the clinical definition of asthma. You can use the GINA definition.
· 4.1., 4.2 and 6. I would suggest summarising the potential clinical benefit of enhancing or blocking IGFBP-6 in a paragraph in each chapter.
· 6. IGFBPs were mostly investigated in lung cancer with contradictory results. Could you please, review the clinical studies in detail and attempt explaining the reasons for discrepancies (i.e. different type, stage, treatment of lung cancer).
Author Response
I have read the review article by Venuto et al with great interest. The authors summarise the knowledge on the role of IGFBP-6 in lung diseases.
Comments:
Q1. Table 1. Asthma. Please, clarify this. Currently, the role of IGFBP-6 is confusing.
A1. In the table we have grouped in a single row the works that concern different aspects of asthmatic pathology, specifying the relative results.
Q2. Figure 1. IGFBP-6 blocks IL1β on the picture, but the description says IL1α. Please, clarify.
A2. It was a typo in the description, which now it has been changed to IL1β.
Q3. 4.1 Please, clarify the clinical definition of asthma. You can use the GINA definition.
A3. We have now used the GINA definition as recommended (yellow-marked in paragraph 4.1).
Q4. 4.1., 4.2 and 6. I would suggest summarising the potential clinical benefit of enhancing or blocking IGFBP-6 in a paragraph in each chapter.
A4. We have now summarized the potential clinical benefit of modulating IGFBP-6 at the end of Subsections as requested by the Reviewer.
Q5. IGFBPs were mostly investigated in lung cancer with contradictory results. Could you please, review the clinical studies in detail and attempt explaining the reasons for discrepancies (i.e. different type, stage, treatment of lung cancer).
A5. We thank the Reviewer for this comment. According to the literature, there is no clear correlation between the expression level of each single IGFBP as an oncogene or tumor suppressor gene product, and tumor type, stage, and treatment. However, to satisfy the request by the reviewer, we added a group of sentences (highlighted in yellow) at the end of Subsection 5.2, in which we speculate that such IGFBP expression differences in lung cancer may occur probably due to the presence of metastases and/or the treatment of the tumor with cisplatin or other inhibitors.
Reviewer 4 Report
With interest, I read the manuscript ijms-2215281.
1. “Normal human inflammatory airway cells possess a powerful array of neurohormones and IGFBPs that are available for modulating local IGF-1 bioavailability in the lung [11]. In this inflammatory scenario, IGFBP-6 is differentially expressed in the lung, in the suppression of allergic airway inflammation, and bronchial biopsies of asthmatic subjects [12, 13] …”. Do the references perfectly match the text? Please, verify. Besides, why this part appears here not in asthma-related section?
2. In continuation, Table 1 with the same question to the references 11 and 12. Besides, what do you mean “allergic airway inflammation” and how it differs from asthma in the referenced papers? Does “allergic airway inflammation” refer to human or animal study here? In general, what do you mean conditions? Diseases? Then also what is “bronchial tissue inflammation”? Finally, how comprehensive is this table? Systematic search?
3. One figure and one Table is not enough for a full review. Could you enhance the graphical representation oof your article?
4. “Concerning the pathobiology of asthma, lung inflammation principally falls into a dichotomy of a preponderant (>50%) type 2-high response -including eosinophilic, allergic, and non-allergic asthma- and a type 2-low response involving a neutrophilic and pauci-granulocytic asthma [51].” Not enough and introducing issue not necessarily clear to each and every reader such as “pauci-granulocytic asthma”. Th1 and -17 involvement should be mentioned (PMID: 30057383, 31904412, 33926084, etc.).
5. The part on MSCs could better fit the asthma section. Please, try to amend it.
6. Generally, the flow between different paragraphs in the sections could be improved.
7. The number of 98 references is not huge. Is it the the topic is emerging and it is therefore? If yes, fine.
Author Response
With interest, I read the manuscript ijms-2215281.
Q1. “Normal human inflammatory airway cells possess a powerful array of neurohormones and IGFBPs that are available for modulating local IGF-1 bioavailability in the lung [11]. In this inflammatory scenario, IGFBP-6 is differentially expressed in the lung, in the suppression of allergic airway inflammation, and bronchial biopsies of asthmatic subjects [12, 13] …”. Do the references perfectly match the text? Please, verify. Besides, why this part appears here not in asthma-related section? –
A1. We would like to thank the Reviewer for having pointed out the appropriateness of references that we have now corrected/amended, as follows:
REF [11] is now correct and refers to Allen, J.T., et al., Expression of growth hormone-releasing factor, growth hormone, insulin-like growth factor-1 and its binding proteins in human lung. Neuropeptides, 2000. 34(2): p. 98-107.
REF [12] is correct and is also mentioned in Table 1.
REF [13] is now Kim, S.D., et al., Screening and Functional Pathway Analysis of Pulmonary Genes Associated with Suppression of Allergic Airway Inflammation by Adipose Stem Cell-Derived Extracellular Vesicles. Stem Cells Int, 2020. 2020: p. 5684250).
This part has been included in the Introduction since we think that it conveys general concepts about different roles of IGF system in various morbidities rather than just in asthma.
Q2. In continuation, Table 1 with the same question to the references 11 and 12. Besides, what do you mean “allergic airway inflammation” and how it differs from asthma in the referenced papers? Does “allergic airway inflammation” refer to human or animal study here? In general, what do you mean conditions? Diseases? Then also what is “bronchial tissue inflammation”? Finally, how comprehensive is this table? Systematic search?
A2. As for the references, please see the answer above.
As to the definition “allergic airway inflammation”, we have reported the exact terms used by the authors in the mice study.
Moreover, in order to provide an easy representation of the various studies, we have combined those works concerning asthma in a single line since they evaluate different aspects (starting tissue, subjects analyzed) of the same pathology.
To better clarify what we mean by “conditions”, we have modified the column title in the table which now refers to “diseases”.
The table has the purpose of summarizing what is reported in the text to immediately provide information regarding the role of IGFBP-6 in the pathologies analyzed. To the best of our knowledge, the cited references are the most appropriate to discuss the role of IGFBP-6 in various lung diseases.
Q3. One figure and one Table is not enough for a full review. Could you enhance the graphical representation of your article?
A3. In keeping with the Reviewer’s request, we have now included a novel Figure, i.e. Figure 1, depicting schematically the IGF/IGF-R/IGFBP system and its role in lung-related topics presented in the Introduction.
Q4. “Concerning the pathobiology of asthma, lung inflammation principally falls into a dichotomy of a preponderant (>50%) type 2-high response -including eosinophilic, allergic, and non-allergic asthma- and a type 2-low response involving a neutrophilic and pauci-granulocytic asthma [51].” Not enough and introducing issue not necessarily clear to each and every reader such as “pauci-granulocytic asthma”. Th1 and -17 involvement should be mentioned (PMID: 30057383, 31904412, 33926084, etc.).
A4. We have added the 3 indicated references (55-57). Also, we have enriched the paragraph by citing the involvement of Th1, Th17 or type 3 innate lymphoid cells in the pathophysiology of asthma (yellow-marked).
Q5. The part on MSCs could better fit the asthma section. Please, try to amend it.
A5. We thank the Reviewer for the valuable suggestion, thus we have revised the text to make it a better fit.
Q6. Generally, the flow between different paragraphs in the sections could be improved.
A6. We have worked out the text to better improve the flow between different paragraphs and changed some of the subheadings.
Q7. The number of 98 references is not huge. Is it the the topic is emerging and it is therefore? If yes, fine. –
A7. The revised version of the manuscript contains now 105 references
Round 2
Reviewer 4 Report
Thank you for addressing my comments well. I have no further reservations.